# Perspectives of Therapeutic Drug Monitoring of Biological Agents in Non-Infectious Uveitis Treatment: A Review

**DOI:** 10.3390/pharmaceutics15030766

**Published:** 2023-02-25

**Authors:** Manuel Busto-Iglesias, Lorena Rodríguez-Martínez, Carmen Antía Rodríguez-Fernández, Jaime González-López, Miguel González-Barcia, Begoña de Domingo, Luis Rodríguez-Rodríguez, Anxo Fernández-Ferreiro, Cristina Mondelo-García

**Affiliations:** 1Pharmacy Department, University Clinical Hospital of Santiago de Compostela (SERGAS), 15706 Santiago de Compostela, Spaincrismondelo1@gmail.com (C.M.-G.); 2Pharmacology Group, Health Research Institute of Santiago de Compostela (FIDIS), 15706 Santiago de Compostela, Spain; 3Ophthalmology Department, Bellvitge University Hospital, 08907 Barcelona, Spain; 4Ophthalmology Department, University Clinical Hospital of Santiago Compostela (SERGAS), 15706 Santiago de Compostela, Spain; 5Musculoskeletal Pathology Group, Hospital Clínico San Carlos, Instituto Investigación Sanitaria San Carlos (IdISSC), 28040 Madrid, Spain

**Keywords:** non-infectious uveitis (NIU), therapeutic drug monitoring (TDM), pharmacokinetics, pharmacogenetics, biological therapy

## Abstract

Biological drugs, especially those targeting anti-tumour necrosis factor α (TNFα) molecule, have revolutionized the treatment of patients with non-infectious uveitis (NIU), a sight-threatening condition characterized by ocular inflammation that can lead to severe vision threatening and blindness. Adalimumab (ADA) and infliximab (IFX), the most widely used anti-TNFα drugs, have led to greater clinical benefits, but a significant fraction of patients with NIU do not respond to these drugs. The therapeutic outcome is closely related to systemic drug levels, which are influenced by several factors such as immunogenicity, concomitant treatment with immunomodulators, and genetic factors. Therapeutic drug monitoring (TDM) of drug and anti-drug antibody (ADAbs) levels is emerging as a resource to optimise biologic therapy by personalising treatment to bring and maintain drug concentration within the therapeutic range, especially in those patients where a clinical response is less than expected. Furthermore, some studies have described different genetic polymorphisms that may act as predictors of response to treatment with anti-TNFα agents in immune-mediated diseases and could be useful in personalising biologic treatment selection. This review is a compilation of the published evidence in NIU and in other immune-mediated diseases that support the usefulness of TDM and pharmacogenetics as a tool to guide clinicians’ treatment decisions leading to better clinical outcomes. In addition, findings from preclinical and clinical studies, assessing the safety and efficacy of intravitreal administration of anti-TNFα agents in NIU are discussed.

## 1. Introduction

Uveitis refers to a heterogeneous group of diseases characterized by inflammation of the uvea, a structure formed by the iris, the choroid, and the ciliary body. They are usually classified depending on their aetiology as infectious or non-infectious, or according to the location of the inflammation (anterior, intermediate, posterior, or panuveitis). Non-infectious uveitis (NIU) has an immune-mediated or idiopathic aetiology and usually occurs in the form of flares [1].

Traditionally, local or systemic treatment with corticosteroids has been the mainstay therapy in patients with NIU. The powerful immunosuppressive effect of corticosteroids makes them highly effective drugs in the control of acute flares; however, long-term treatment can lead to the appearance of adverse effects or other ocular complications, especially in patients with chronic doses of prednisone equivalents over 7.5 mg per day [2]. Therefore, in many cases, it is necessary to associate other immunomodulators that allow for the reduction of the long-term adverse effects of corticosteroids while enhancing their immunosuppressive action [3]. The immunomodulatory drugs commonly used as first-line treatment in NIU are antimetabolites such as methotrexate (MTX), mycophenolate mofetil (MMF), or azathioprine (AZA), calcineurin inhibitors such as cyclosporine, or alkylating agents such as cyclophosphamide. These drugs are also known as “corticosteroid-sparing agents” since corticosteroid doses can be reduced while maintaining good control of ocular inflammation. Although treatment with immunosuppressants has led to substantial improvement in the management of NIU [3], in some cases the ocular inflammation persists. However, biological drugs have emerged in recent years as useful resources in many forms of NIU that do not respond to conventional treatment [4,5].

Numerous studies have confirmed the favourable results both in efficacy and safety of biological drugs, mainly molecules against tumour necrosis factor α (TNFα), in patients with uveitis refractory to conventional treatments. The introduction of these drugs in the treatment of NIU has been possible due to the knowledge gained about the inflammatory mediators involved in this pathology, thus allowing the use of drugs that act specifically against these molecules [4]. Despite the wide use of various biological drugs including anti-TNFα drugs for the treatment of autoimmune diseases, only adalimumab (ADA) has been approved by the Food and Drug Administration (FDA) and the European Medicines Agency (EMA) (2014 and 2016, respectively) in non-anterior NIU [6]. After the introduction of ADA for the treatment of NIU, better control of inflammation has been achieved, with improvements in visual quality and fewer complications, but there is still a high percentage of patients, around 40%, who present primary treatment failure in the first six months according to the results of VISUAL clinical trials [6,7]. However, results from a recent real-world data study showed higher drug retention rates close to 55% after the first five years, with inefficacy being the main cause of discontinuation [8]. In non-responders, treatment with ADA is not only not beneficial, but it can produce undesired adverse events. This situation highlights the need to identify factors related to treatment response with biological drugs that allow better management of patients with NIU. In this sense, the study of pharmacokinetic (PK) and pharmacogenetic (PG) parameters related to anti-TNFα drugs appears to be promising in the identification of biomarkers for treatment response.

Studies that have evaluated factors related to failure in therapy with anti-TNFα agents in the field of ophthalmology are scarce in comparison with those in immune-mediated rheumatic and gastrointestinal pathologies. This is mainly due to the short course of anti-TNFα drugs in the treatment of NIU, although the low prevalence and high heterogeneity of the disease also contribute. PK studies in other immune-mediated pathologies have revealed a significant inter-individual variability in the systemic concentrations of biological drugs. This has been related to the development of anti-drug antibodies (ADAbs), the use of concomitant immunomodulatory treatments, and alterations in biochemical parameters, among which albumin is one of the most important [9]. This variability together with the difficulty of access of the drug to the site of action may be two relevant aspects of conditioning response to treatment. From a biopharmaceutical point of view, the eye has barriers that limit the passage of high molecular weight molecules. However, the presence of inflammation can facilitate the passage of large molecules such as biologic drugs [10]. The passage of monoclonal antibodies (mAbs) through the ocular barriers toward the eye is a barely studied aspect, but it could have a great impact on the pharmacodynamic aspects of these drugs regarding the treatment of NIU. On the other hand, PG studies in immune-mediated pathologies have provided evidence of the influence of certain genetic polymorphisms in the response to biological drugs, although the relevance of these findings in NIU is currently unknown. Thus far few studies have been conducted aimed at evaluating the clinical relevance of PK and PG aspects in NIU.

Compared with classical drugs such as digoxin, mAbs used as therapeutic proteins have different pharmacokinetic characteristics due to their particular physical characteristics, generally showing a smaller volume of distribution and a longer half-life. Classic drugs are eliminated through hepatic metabolism, renal filtration, and excretion through bile or faeces. In contrast, the clearance of therapeutic antibodies is related to protein degradation and target binding. This phenomenon known as target-mediated drug disposition (TMDD), is also present in small molecules that exhibit non-linear pharmacokinetics, although it is more frequent in therapeutic proteins [11]. Although the pharmacokinetics of classical drugs and therapeutic proteins differ, in both treatment scenarios systemic drug concentrations falling within a specific therapeutic range should be achieved to maximize treatment efficacy and clinical outcomes while avoiding undesired adverse events that may arise from excessively high drug levels. However, in clinical practice, this is not achieved in all patients due to the inter-individual variability in treatment, which highlights the need for tailored therapy. Therapeutic Drug Monitoring (TDM) is a useful tool to meet this need with which treatment can be personalized to reach therapeutic concentrations.

The introduction of biological therapies has revolutionized the treatment of NIU, especially those targeting TNFα. However, NIU treatment is challenging and current strategies are sometimes insufficient to achieve adequate control of ocular inflammation. Approximately 40% of patients experience early treatment failure (primary non-response, PNR), whereas up to 20% of patients experience an initial clinical improvement followed by a loss of response 12 months after starting treatment (secondary non-response, SNR) [6]. Given the limited repertoire of effective biologic drugs available in NIU, early identification of non-response (PNR) or loss of response (SNR) is of utmost importance in clinical practice.

The therapeutic outcome of biologic drugs in immune-mediated diseases is closely related to serum drug concentration [12,13]. Whilst therapeutic failure to anti-TNFα agents is commonly associated with low or undetectable serum trough drug levels (subtherapeutic), therapeutic success is associated with trough drug levels over a specific threshold, in a range in which maximum favourable outcomes are achieved with minimal or no adverse events [14]. Hence, the implementation of therapeutic drug monitoring (TDM), consisting of measuring trough drug levels and ADAbs, is essential to assess in each patient the performance of a given biological drug and define its optimal dose ranges. 

Several approaches are currently available to measure drug and anti-drug antibody concentrations. Enzyme-linked immunosorbent assay (ELISA)-based techniques and radioimmunoassay (RIA) are the most used tests compared to other assays such as Homogenous Mobility Shift Assay (HMSA) and immunological multiparameter chip technology (IMPACT) [15,16]. Each assay format has a different sensitivity, dynamic range, and cut-points, so the results obtained are not equivalent. Therefore, the method of choice to monitor drug levels and anti-drug antibodies should be reported in all studies. All assays have advantages and limitations, some of them are inherent to the specific methodology, but others are related to economic factors, the presence of adequate facilities, or qualified personnel, among others. A detailed description of the available assays, their characteristics, and their main benefits and limitations can be found elsewhere [15,16].

Ideally, anti-TNFα therapy should result in therapeutic concentrations in all patients, but this is not always achieved in clinical practice as evidenced by the large differences observed in the systemic concentration of anti-TNFα drugs [17,18,19,20]. These differences are likely explained by heterogeneous drug bioavailability in patients, which in turn is influenced by PK factors, such as drug immunogenicity [21].

Furthermore, TDM was typically considered advantageous for drugs with a large inter-individual variability in exposure with relatively low intra-individual variation, a significant exposure–efficacy relationship, a narrow therapeutic window, and the availability of a validated bioanalytical assay. It has been postulated recently that this could also represent a useful tool to individualize dosing and optimize treatment using drugs with a wide therapeutic window and high cost [22].

The purpose of this comprehensive review is to compile the available evidence on the PK, TDM, and PG monitoring of anti-TNFα drugs used in NIU, as well as to discuss the relevance of the biopharmaceutical considerations that concern drug delivery in the eye, in relation to biological drug treatment. In addition, other routes of administration used for the administration of anti-TNF, such as the intravitreal route, will be mentioned. For that purpose, an extensive review of the preclinical and clinical pharmacokinetic studies that have been published in this field was carried out. The PubMed database was searched from its inception in 2000 to September 2022, and the reference lists from the relevant studies were analysed for additional literature.

## 2. Biologics in Uveitis Treatment

A better understanding of ocular inflammation pathways has led to the emergence of biological therapies for the treatment of NIU [3,23] that aim to overcome the 30% failure rate obtained under classical immunosuppressive treatment [24]. Different cytokines such as TNFα, IL-6, IL-17, or IL-23 play a key role in NIU inflammatory process [25], therefore becoming very attractive as potential therapeutic targets. Bearing this in mind, randomized prospective studies have been developed in the last years to evaluate the treatment efficacy of biological drugs in NIU [6,7,26]. In this sense, the main biological drugs used in NIU treatment are depicted in Table 1.

Adalimumab: ADA is a fully human monoclonal antibody targeting TNFα, which plays a central role in ocular inflammation via reactive oxygen species, inducing angiogenesis and breakdown of blood–retinal barrier (BRB). The time to reach maximum serum concentration is 56 h, after a 40-mg subcutaneous administration to a healthy adult subject, with an average absolute bioavailability estimated at 64%. The mean terminal half-life is approximately two weeks [27]. The first report about the role of ADA in NIU was in 2008 [28], after which several studies established the effectiveness of ADA in NIU, mainly the VISUAL I and VISUAL II trials. Both were randomised and multicentric clinical trials compared with a placebo, which evaluated the efficacy and safety of ADA in active and inactive NIU [6,7]. Based on the findings of these trials, ADA was approved for its use in NIU and is currently the only biological treatment approved by FDA and EMA for this purpose [1]. Phase III extension study (VISUAL III) has shown that ADA treatment maintains disease control and provides long-term corticosteroid-sparing effects [29]. In addition, a recent meta-analysis of six randomized controlled trials (RCTs) evaluating the efficacy and safety of ADA treatment of NIU has shown that treatment failure is halved compared with placebo, as well as a reduction in visual loss and ocular inflammation [30]. Future research aims to directly compare the efficacy of ADA in monotherapy and in combination with other immunosuppressants [31].

Infliximab (IFX): IFX is a chimeric (human/mouse) monoclonal antibody targeting TNFα has a half-life of up to 9.5 days and is administered intravenously. IFX use in uveitis was first reported in 2001 [32] and has been shown to be effective for NIU in children [33] and for Behçet’s disease-associated uveitis resistant to classical immunosuppressive treatment [34], although it may also be effective for the management of other ocular diseases [1,4]. The use of IFX for the treatment of patients with refractory uveoretinitis of Behçet’s disease (RUBD) has been approved in Japan in 2007 [35]. Its early use is strongly recommended in patients with vision-threatening ocular manifestations of Behçet’s disease and should be considered as second-line therapy in juvenile idiopathic arthritis (JIA) related uveitis [36], proven its efficacy and its safety at doses as high as 20 mg/kg successfully used in these patients [37]. Furthermore, comparable results in terms of efficacy have been reported between IFX and ADA treatments for NIU [38]. 

Etanercept: Etanercept is a human recombinant fusion protein consisting of the ligand-binding region of the TNF-R2 receptor coupled to the constant region of immunoglobulin G1 (IgG1-Fc), which inhibits the attachment of TNFα to endogenous TNF receptors. Its half-life is around 70 h. Etanercept is approved for use in rheumatoid arthritis (RA), psoriatic arthritis (PsA), plaque psoriasis (PS), ankylosing spondylitis, and polyarticular JIA, whereas its use in NIU is limited to case reports and small clinical trials [4]. Paradoxically, a significant association between etanercept and the development of uveitis as a drug-associated side effect has been reported compared to ADA or IFX [39]. Therefore, it is strongly recommended that the use of either of these two anti-TNFα agents should be considered before etanercept therapy for the treatment of ocular inflammatory disease [36].

Golimumab: Golimumab, a fully human monoclonal antibody targeting TNFα, with a half-life of about 12 days, has shown potential efficacy in patients with refractory NIU to TNFα blockers [40,41], emerging as a promising therapeutic option in this disease. Nevertheless, all data were obtained from retrospective case series with small sample sizes, so additional studies on its efficacy are required. 

Certolizumab: Certolizumab is a PEGylated antigen-binding fragment (Fab’) of a recombinant humanized monoclonal antibody to TNFα. The conjugation of the hydrophilic polyethylene glycol (PEG) chains increases the half-life of certolizumab pegol to around two weeks. The clearance of certolizumab differs from that of other biological agents due to the absence of an fc fragment in its structure, which prevents FcRn-mediated recycling. In addition, renal excretion of certolizumab has been described due to the relatively small size of the Fab’ fragments [42,43]. Data on the efficacy of certolizumab in the treatment of NIU are limited to case series showing it may be effective in the inflammatory control of refractory NIU [44].

Tocilizumab: Tocilizumab is a humanized monoclonal antibody that inhibits IL-6 signalling by preventing IL-6 from binding to its receptor. A prospective randomized trial evaluated tocilizumab safety and efficacy for the treatment of non-anterior uveitis and observed significant improvement in visual acuity and a reduction of central foveolar thickness [45]. Additionally, tocilizumab has demonstrated efficacy in managing JIA-associated uveitis refractory to anti-TNFα therapy [46], Behçet-associated uveitis [47], birdshot chorioretinopathy [48], and uveitic macular oedema [49].

Rituximab: Rituximab is a B-cell-depleting chimeric anti-CD20 monoclonal antibody. The mean terminal half-life is approximately 22 days. A growing number of reports have supported the use of rituximab in some types of NIU [50,51,52,53,54,55]. A retrospective study in JIA-associated uveitis showed a decrease in uveitis recurrences in patients who have not previously responded to other biologic therapies [50]. Additionally, rituximab treatment resulted in clinical improvement in 14 patients with Vogt–Koyanagi–Harada (VKH) disease-associated uveitis [51,52], 20 patients with severe manifestations of Behçet-associated uveitis [53] and induced remission in 20 patients with refractory ophthalmic Wegener’s granulomatosis [54].

**Table 1 pharmaceutics-15-00766-t001:** Monoclonal antibodies used in different types of NIU.

Drug	Target	Structure	Dosage	Uveitis Type	References
Adalimumab ^1^	TNF-α	mAb, fully humanized	LD: 80 mg MD: 40 mg every other week	Non-infectious non-anterior uveitis ^†^	[6,7]
Infliximab	TNF-α	mAb, mouse-human chimeric	LD: 5 mg/kg at weeks 0, 4, and 6 MD: 5 mg/kg every 4 to 8 weeks Max. dose: 10mg/kg for adults, 20 mg/kg for children every 4 weeks	JIA-related uveitis, Behçet, VKH, sarcoidosis, pars planitis, birdshot retinochoroidopathy, idiopatic uveitis	[1,4,33,34,37]
Etanercept ^2^	TNF-α	Human fusion protein	50 mg weekly	Behçet	[4]
Golimumab	TNF-α	mAb, fully humanized	MD: 50 mg monthly Max. dose: 100 mg monthly	Refractory uveitis	[40,41]
Certolizumab	TNF-α	mAb, fully humanized	200 mg every 2 weeks	Refractory uveitis	[44]
Tocilizumab	IL-6	mAb, fully humanized	4–12 mg/kg every 2–4 weeks	Non-infectious non-anterior uveitis, Behçet, birdshot, JIA-related uveitis	[46,47,48,49]
Rituximab	CD-20	mAb, mouse-human chimeric	LD: 500 or 1000 mg at 0, and 2 weeks. MD: Repeat at 6–12 months if needed	Refractory uveitis, JIA-related uveitis, Behçet, VKH, Wegener’s granulomatosis	[50,51,52,53,54,55]

^1^ Only biological drug indicated to treat non-infectious intermediate, posterior, and panuveitis. ^2^ Use of infliximab or adalimumab should be considered before etanercept therapy for the treatment of ocular inflammatory disease. ^†^ Adalimumab on-label indication. LD: loading dose, MD: maintenance dose, TNFα: tumour necrosis factor-α, IL-6: interleukin-6, CD-20: cluster of differentiation-20, mAb: monoclonal antibody.

## 3. Therapeutic Drug Monitoring of Anti-TNFα in NIU

### 3.1. Pharmacokinetics (PK) of mAbs

MAbs are glycoproteins based on the structure of γ-immunoglobulins (IgG); hence they have a high molecular weight. These drugs are administered parenterally, either intravenously (IV), subcutaneously (SC), or intramuscularly (IM). Due to their high molecular weight, mAbs are absorbed through the lymphatic system after SC or IM administration. The high molecular weight of mAbs also hinders their distribution to tissues and therefore, they are retained in vascular and interstitial spaces. Consequently, these drugs usually have small volumes of distribution [56]. mAbs are protected from lysosomal degradation by the neonatal Fc receptor (FcRn), located in a wide variety of tissues throughout the body, which explains their long half-life and low clearance [57]. The main elimination pathway of mAbs is proteolytic degradation, in contrast to low molecular weight drugs, which are usually eliminated by renal or biliary excretion or by metabolic biotransformation [56]. Antigen mass, which refers to the total amount of antigen available for mAb binding, also influences the PK of mAbs. An increase in antigen mass has been related to an increase in mAb clearance. In other words, in the presence of high antigen amounts most of the mAb molecules form antigen-antibody complexes rather than remain as free antibodies. The elimination rate of these complexes is faster than that of the free mAbs, which explains the increased clearance [58].

PK of mAbs is highly variable. A clarifying example is the inter-individual variability in the clearance of some mAbs used in RA, quantified between 17 and 44% [43]. Multiple cofactors can act as sources of this variability (Figure 1), among which the development of immunogenicity stands out [42]. This eventually translates into variability in the concentration of the mAbs, which markedly influences the therapeutic response. Accordingly, a more extensive review of the PK of mAbs is available elsewhere [21]. The present review will focus on the PK of mAbs used in the treatment of NIU.

#### 3.1.1. Demographic Factors

Body size and gender have an impact on the PK of mAbs and other biological drugs. An increase of clearance with body weight or body surface area has been reported for ADA [59,60,61] and IFX [62], but also for rituximab [63], etanercept [64], and golimumab [65]. Clearance of ADA and IFX is significantly higher in men than in women [59,60], although this may be explained by differences in body weight between gender. Despite the direct relationship between ADA and IFX clearance with body size, ADA dosage is not adjusted to weight in the adult population for the treatment of NIU, whereas IFX doses are weight-adjusted. This fact is linked to the subcutaneous administration of ADA by the patient, which limits dose adjustment by weight, in contrast to the extemporaneous preparation of IFX, enabling individualised weight adjustment at each administration. Moreover, although ADA clearance is increased in heavier patients, in VISUAL I and VISUAL II subgroup analysis by weight, ADA was favoured in all weight subgroups with standard dose [61].

#### 3.1.2. Biochemical Factors

In inflammatory bowel disease (IBD), an inverse relationship between serum albumin levels and clearance of ADA [66] and IFX [67] was found. The reasoning behind this relationship is that low albumin levels may reflect decreased FcRn activity and thus increased clearance [43,57]. C-reactive protein (CRP) also influences the PK of IFX. Specifically, a direct correlation has been described between CRP levels and IFX clearance [68]. 

#### 3.1.3. Immunogenicity of mAbs

Biological drugs are exogenous proteins, so they can induce an immune response. The development of ADAbs was more frequent in patients treated with IFX (ADAbs+: 25.3% (CI 19.5–32.2)) compared to those receiving other biological drugs (ADAbs+: <14%) according to a meta-analysis conducted in more than 14,000 patients with RA, IBD, and spondyloarthritis (SpA) [69]. The appearance of immunogenicity against the drug can lead to undesired issues such as loss of response, as well as the development of severe adverse infusion reactions [42,69]. Additionally, immunogenicity has a great impact on the PK of mAbs by increasing their clearance. In patients with NIU treated with ADA, an approximately 3-fold increase in clearance has been reported in those who developed ADAbs compared to those without ADAbs [61]. This finding is consistent with the decrease in serum levels of ADA and IFX [17,18,61,70] and with the worse clinical response observed in patients with NIU who present ADAbs [18,70] and is also in line with previously described findings in patients with immune-mediated rheumatic and gastrointestinal diseases [13,71,72,73,74].

#### 3.1.4. Concomitant Immunosuppressive Therapy

Several studies have shown that the beneficial effect of anti-TNFα drugs is enhanced by concomitant treatment with immunosuppressants [75,76,77,78,79,80]. Although other studies have not observed any additional effect over monotherapy with biological agents [81,82], many clinical practice guidelines recommend the combined treatment of immunosuppressants and biologic drugs, in order to improve the pharmacokinetics of the biologic agent (increasing trough concentration and decreasing immunogenicity) [83]. This combined therapeutic strategy has proven superior to monotherapy treatment with ADA [75,76,77] and with other biological drugs [76,78,79,80] in patients with other immune-mediated pathologies. The superiority of combined therapy over monotherapy is reflected by significantly higher response rates or higher remission rates, improvements in inflammatory and disease activity parameters, less treatment failure or less damage to the affected tissues, and other clinical measures, without producing a higher frequency of adverse effects [75,78,79,80]. Potentiation of the therapeutic effect exerted by the co-administration of immunosuppressants is attributed to the higher concentration of the biological drug detected in blood compared to the absence of immunosuppressant treatment [77,84]. The association detected between co-treatment with immunosuppressants and decreased drug clearance likely explains these findings. Specifically, the decrease in ADA clearance with concomitant immunosuppressant treatment has been estimated at 38.4% in patients with NIU [61]. It should be noted that co-treatment with immunosuppressants reduces the risk of developing immunogenicity, as well as its associated negative effects [73,77,85], which has a direct impact on the clearance of biological drugs, as already mentioned. Altogether, these findings indicate that concomitant treatment with immunosuppressants has a protective effect against the appearance of immunogenicity, which results in an increase in systemic drug levels and therefore a higher probability of clinical response.

### 3.2. Evidence Supporting TDM of Anti-TNFα in NIU

The inter-individual variability introduced by these and other factors in the PK parameters of anti-TNFα drugs has an impact on the individual exposure to the drug and therefore on the clinical response [21]. Evidence supporting TDM of drug levels and ADAbs of TNFα inhibitors as an effective strategy to optimize biological therapy and increase treatment response rates in immune-mediated inflammatory diseases is growing [12,14,17,73,74,86,87,88]. However, opposing findings have also been reported [89,90,91]. Given the shared similarities between these diseases and NIU regarding anti-TNFα therapy, it is reasonable to assume that TDM may also be beneficial in the management of patients with NIU. In fact, several publications support this idea, as shown in Table 2. Most of the data have been obtained from works published in the literature and have been completed with a report from the EMA.

Significantly higher trough ADA levels have been consistently reported in patients with NIU who responded to treatment compared to those non-responders. In addition, ADA trough levels have shown an inverse correlation with anti-ADA antibody levels [17,18,61,70,92]. Consequently, the presence of anti-ADA antibodies has been associated with treatment failure as assessed by a worse uveitis outcome and failure to achieve remission. This association was evaluated in more detail in two studies in which anti-ADA antibodies were classified as permanent if positive results were obtained on two or more time points during follow-up or transient if obtained on one single occasion [17,18]. In both studies, ADA trough levels were undetectable in patients with permanent anti-ADA antibodies, but not in those with transient antibodies, whose ADA trough levels did not differ from that in seronegative patients. Therefore, an inverse correlation of ADA trough levels and anti-ADA antibody levels was only observed when permanent antibodies were detected. Moreover, the presence of permanent antibodies was associated with a worse uveitis outcome and increased likelihood of non-response. In the study of Skrabl-Baumgartner et al. [17] 77.8% of non-responders developed permanent anti-ADA antibodies. In contrast, such an association was not observed in patients with transient antibodies. The authors concluded that the development of immunogenicity was the main reason for the loss of response, although other variables were not analysed. Although these results should be interpreted with caution due to the low number of patients with permanent anti-ADA antibodies analysed, which were 4 in Cordero-Coma et al. [18] and 7 in Skrabl-Baumgartner et al. [17], they derive from independent studies with different collections of patients and likely represent true rather than fortuitous findings.

Assuming this is the case, it would be of great importance to monitor ADAbs levels throughout treatment to differentiate between transient and permanent antibody positivity, since only the latter is associated with subtherapeutic drug levels and with a higher risk of treatment failure. Subsequent studies have confirmed the relationship between a worse clinical response with lower trough ADA levels, which in turn are frequently found in patients with ADAbs [17,61,70,92]. Despite most studies in NIU having shown a protective effect of immunomodulatory therapy against the development of immunogenicity [17,61,70] in accordance with previous reports in other pathologies, Cordero-Coma et al. [18] did not observe such an effect. In the work performed by Sugita et al. [93], monitoring of IFX levels showed a tendency towards higher IFX concentration in patients without uveitis flares (responders) and higher rates of treatment response in those with levels over 1 ug/mL (Table 2), results that are in line to those observed with ADA treatment. Convincing evidence that TDM-guided optimization of ADA therapy in patients with NIU results in relevant clinical improvements has been recently reported by Sejournet et al. [92]. The authors showed that treatment adjustment in non-responders (increase in injections, dose, or change in treatment) according to TDM results, led to clinical improvement in 87% of cases, while in responders with supratherapeutic ADA levels, the reduction in the number of injections did not lead to relapse in 80% of cases (Table 2). 

A question that remains to be answered is the potential utility and cost-effectiveness of proactive versus reactive TDM in the management of patients with NUI. Results obtained in IBD show greater clinical benefits and lower costs for proactive TDM compared to reactive TDM [86,94,95], probably because the proactive approach allows early intervention before a loss of response occurs and detection of immunogenicity early in treatment [96]. The potential benefits of proactive TDM of anti-TNF inhibitors in NIU need to be specifically evaluated in these patients. However, before reaching that point, some important issues remain to be clarified, such as the therapeutic range of anti-TNF drugs in NUI or the optimal frequency for monitoring [96].

Furthermore, there are reference documents on certain types of NIU that recommend using TDM to guide treatment with biologicals, such as the guide developed by the Single Hub and Access point for paediatric Rheumatology in Europe (SHARE) initiative. This document contains recommendations for the management of patients with JIA-associated uveitis [97], which indicates that increasing the dose or shortening the interval of drug administration can be considered in non-responders who do not have ADAbs but have low drug levels. Expert recommendations as well as an algorithm for the treatment of JIA-associated uveitis have also been published, which recommends adjusting treatment with ADA and IFX in non-responders or in cases of loss of response based on the results of monitoring levels of drug and ADAbs [98]. 

**Table 2 pharmaceutics-15-00766-t002:** Relationship between anti-TNFα drug levels with ADAbs development, concomitant DMARDs, and response in NIU patients.

Authors [Ref.]	Type of Study	Treatment	Type NIU	No. Patients	Results
Cordero-Coma et al. 2016 [18]	Observational, prospective	ADA	Refractory uveitis	25 Naïve to biologics	- ADA levels: 0.6 µg/mL in non-responders vs. 9.5 µg/mL in responders (*p* < 0.001); 11.8 µg/mL in complete responders vs. 8.6 µg/mL in partial responders (*p* = ns). - Levels of permanent AAA (n = 4) inversely correlated with ADA levels (*p* < 0.001). - Presence of permanent AAA (n = 4) associated with worse clinical evolution of uveitis (*p* = 0.014).
Skrabl-Baumgartner et al. 2019 [17]	Observational, prospective	ADA	JIA related uveitis	20	- ADA trough levels inversely correlated with AAA levels in patients with permanent AAA (*p* < 0.001). - Transient AAA was not correlated with a reduction in the response. - Significantly lower use of immunosuppressants in patients with permanent AAA (*p* < 0.05).
Leinonen et al. 2017 [70]	Observational, retrospective	ADA	JIA related uveitis	31	- AAA levels ≥ 12 AU/mL were associated with a higher grade of uveitis (*p* < 0.001), lower ADA levels (*p* < 0,001), and lack of concomitant MTX therapy (*p* = 0.043).
Sejournet et al. 2021 [92]	Observational, retrospective	ADA	JIA related uveitis	79	- Significantly higher ADA levels in responders than in non-responders (*p* = 0.0004). - In 24/31 cases of therapeutic adjustment in non-responders, an improvement was observed in 87% of cases.
EMA/501143/2016 [61]	Phase III studies (VISUAL I and VISUAL II)	ADA	NIU	249 (118 VISUAL I/131 VISUAL II)	- Responders had slightly higher ADA levels than non-responders after week 8. - Patients with AAA had lower ADA serum levels than those without AAA.
Sugita et al. 2011 [93]	Observational, prospective	IFX	RUBD	20	- IFX levels: 3.4 µg/mL in patients with uveitis flare (non-responders) vs. 7.3 µg/mL in patients without flare (responders), (*p* = ns). - Response to IFX (no development of uveitis flare): 14/16 (87.5%) with >1 µg/mL IFX vs. 1/4 (25%) with < 1 µg/mL IFX.

ADA: adalimumab, IFX: infliximab, JIA: Juvenile Idiopathic Arthritis, NIU: non-infectious uveitis, RUBD: refractory uveoretinitis of Behçet’s disease, AAA: anti-adalimumab antibodies, ns: non-significant.

These studies show that anti-TNFα trough levels were higher in patients who responded to treatment compared to non-responders, in addition, demonstrate that ADAbs development was associated with worse NIU outcomes, and showing that treatment adjustment according to TDM results led to clinical improvement in non-responders [18,70,74,92,93]. All these data, together with expert recommendations and other reference documents, support the use of the biologic drug TDM in NIU. Despite this, the observational design of these studies added to the small size of the population studied, as well as the heterogeneity of the included NIU limits generalizations. Although we believe the data are in line with observations in other pathologies [12,14,86,87,88] and constitute a step forward in the difficult daily management of refractory NIU patients, they are insufficient to implement TDM in routine clinical practice. 

### 3.3. TDM-Based Strategies and Therapeutic trough Levels of Anti-TNFα in NIU

Overall, the aforementioned studies indicate that treatment with anti-TNFα drugs could be optimized with the implementation of TDM in the treatment decision-making process. However, it must be considered that the strength of the evidence obtained in patients with NIU is lower than in patients with other immune-mediated diseases in which the utility of TDM of biological drugs has been studied more thoroughly. The importance of optimizing biological treatment to a maximum is even more remarkable in patients with NIU who do not show a sufficient response or do not respond to ADA or IFX, in whom effective therapeutic options are even more scarce [34]. In order to prevent PNR or SNR to anti-TNFα drugs, different strategies can be adopted depending on TDM-based trough drug levels and the presence of immunogenicity. In non-responders or poor responders with low trough drug levels who do not develop immunogenicity, dose increase or interval shortening is recommended to increase drug concentration, but if immunogenicity is present, the addition of an immunomodulator or change to another anti-TNFα is a viable option. If the response to anti-TNFα is inadequate despite enough drug levels, a change to another biological drug with a different target is recommended instead [99]. Treatment should be continuously adjusted with subsequent drug concentration reappraisal until reaching the therapeutic target.

The minimum trough anti-TNFα levels that are associated with clinical response in NIU are currently unknown. The identification of a therapeutic range for anti-TNFα drugs is hampered by variability between studies and methodologies used to measure drug levels and ADAbs that generates incomparable data [100]. Prior to the implementation of TDM of ADA (or other biological drugs), a robust description of ADA PK and pharmacokinetic–pharmacodynamic (PK-PD) relationship should be obtained in patients with NIU [59], as an isolated measure of the systemic trough drug concentration does not fully explain drug exposure in a patient. The individual pharmacokinetic profile provides more accurate information on the actual systemic exposure of the drug instead. This profile is obtained by estimating the drug concentration curve (PK curve) as a function of time in each patient from measurements of the systemic trough drug levels from various samples obtained throughout the treatment. The relationship between the PK profile and the response is better known in other drugs such as antibiotics. In the case of beta-lactams, the longer their concentration (area under the curve that reflects the plasma levels of the antibiotic) is maintained above the minimum inhibitory concentration (MIC), the better response is achieved, and therefore they are considered time-dependent [101,102]. In other drugs such as aminoglycosides, in contrast, their effectiveness depends on reaching a sufficient level of maximum plasma concentration (Cmax) with respect to the MIC, and consequently, they are considered concentration-dependent [103]. In the case of anti-TNFα drugs, the influence of the PK profile on the response is unknown, i.e., it is not known whether they can be considered time- or concentration-dependent. Despite this, different dosage schedules are used in clinical practice regardless of the anti-TNFα drug to intensify treatment as discussed above [104]. However, some of these strategies may be ineffective, since the PK profile of the anti-TNF drugs determines the concentrations reached and the time they remain above a minimum value required for therapeutic action at the site of inflammation. This is especially important when the target of anti-TNFα drugs is in anatomical places that are difficult to access, such as the eye.

To our knowledge, an ADA concentration–effect relationship using PK–PD modelling of data from VISUAL I and VISUAL II phase III studies has been only described in one EMA report [61]. The mean steady-state serum ADA concentrations observed in the combined analysis of the two studies were 8–10 µg/mL, identical to that observed in patients with CD, UC, RA, and PS under the same initial and maintenance dose [61]. Strikingly, this concentration range was close to the lower level needed to prevent treatment failure based on estimated mean half maximal inhibitory concentration (IC50) values of 9.7 μg/mL (95% CI 5.5–17.4 μg/mL) and 6.4 μg/mL (95% CI 3.8–10.8 μg/mL) in VISUAL I and VISUAL II studies, respectively. Conversely, in patients with rheumatic diseases, the serum ADA concentration range associated with clinical response is considerably lower, 2–8 µg/mL in RA 2.5–8.0 µg/mL in SpA, and 1–8 µg/mL in PsA [14,84,105]. The scenario is more complex in IBD, as reflected by the broader therapeutic thresholds (3.7- > 12 μg/mL) reported for ADA [14]. Reasonably, different optimal threshold concentrations of ADA are expected to emerge for NIU than for other immune-mediated diseases depending on disease severity, underlying inflammatory mechanisms, the target site of action, and other patient characteristics. These differences in systemic ADA levels associated with treatment response between pathologies are likely related to the heterogeneous bioavailability of ADA across different tissues. 

Assuming that suboptimal steady-state ADA concentrations are achieved with the current standard doses prescribed in NIU (80 mg loading dose, followed by 40 mg every other week) [61], it is worth considering using a higher loading dose of ADA or increasing the dosing frequency to shift towards therapeutic concentrations. The former approach has already proven beneficial in Crohn’s disease [19] and may also be beneficial in RA based on simulations [59]. In addition, simulations of a maintenance dose of 40 mg every week in NIU suggested that treatment failure could be reduced by up to 15% compared to the standard maintenance regimen [61]. Future studies should address whether the administration of intensified dose regiments in patients with NIU at the initiation of ADA therapy will lead to meaningful clinical benefits compared to the standard dose regimen without substantially increasing the frequency and severity of adverse events, thereby reducing the non-negligible rate of PNR and SNR. However, it should be noted that this intensification strategy may not be beneficial in some cases, for example when detectable levels of ADA are present at the time of treatment failure, indicating that molecules other than TNFα are acting as drivers of the inflammatory response. This reinforces the idea that TDM could help to individually optimise therapy to avoid unnecessary overtreatment and related adverse events and costs expenditures [86,87]. 

## 4. Implications of Ocular Drug PK 

As already mentioned, target levels of anti-TNFα drugs are expected to vary among immune-mediated pathologies due, at least in part, to differences in PK parameters and biodistribution of biological drugs across ocular, intestine, and synovial tissues. The eye resides behind particularly strong blood–tissue barriers formed by endothelial-cell tight junctions and other structural specializations that selectively control the transport of molecules and have a great impact on the ocular bioavailability of the anti-TNFα drugs [10]. 

Anti-TNFα systemic administration: following a systemic administration, drugs can reach the choroid and then travel from the blood circulation to the ocular cavity. This process is controlled by two major barriers: the blood–aqueous barrier (BAB) and BRB, located at anterior and posterior segments respectively [106]. These barriers limit drug penetration from the blood into the eye, thus reducing its bioavailability in the target site of action [107]. Non-fenestrated endothelium of the iris vessels and the non-pigmented epithelium of the ciliary body are the main components of BAB. However, the barrier functionality of BAB is not complete, capillaries of the ciliary are fenestrated and leaky to macromolecules, allowing them to reach the aqueous humour (Figure 2). BRB consists of two types of cells, including retinal capillary endothelial and retinal pigment epithelium cells [108]. Consequently, after systemic administration of anti-TNFα, the intraocular concentration is lower than the blood concentration and therefore, patients with NIU may require elevated systemic anti-TNFα trough levels to increase intraocular bioavailability and reach a therapeutic effect.

Only a few preclinical studies have characterized the ocular PK of systemically administered mAbs, whereas ocular PK following intravitreal administration has been well described [109]. The ocular PK of rabbit Fab’ fragments (rabFab, 48 kDa) after systemic administration differs from that of rabbit IgG (rabIgG, 150 kDa) [110]. In aqueous humour and vitreous humour, rabFab showed a fast absorption phase (Tmax 0.5 days) followed by a rapid decline. In contrast, rabIgG showed a relatively slow absorption (Tmax 1–4 days) followed by a slow decline. This is consistent with the accelerated clearance suffered by Fab proteins lacking an Fc fragment, whereas full-length antibodies and fusion proteins with Fc fragments have longer half-lives. Despite rabFab showing higher relative exposure between aqueous humour/serum and vitreous humour/serum and higher percent ocular partition compared to rabIgG, absolute exposure in ocular compartments was higher for rabIgG [110]. These differences may be relevant when estimating the ocular distribution of biological drugs with different structures such as ADA or IFX (IgG structure mAb) and certrolizumab (PEGylated fragment of a humanized mAb). Although informative, these parameters are derived from preclinical studies carried out under non-inflammatory conditions, and with antibodies whose pharmacological targets were not present in the animals studied. The presence of inflammation can alter the biodistribution coefficients at the tissue level. In fact, patients with uveitis experience an increase in vascular permeability due to the release of inflammatory mediators, such as TNFα, IL-6, IL-8, IL-17, or IL-23 [25,111,112]. This increase in vascular permeability has an impact on the integrity of the ocular barriers, which may play an important role in the penetration of high molecular weight drugs such as mAb in tissues with limited access like the eye [113]. Therefore, further preclinical studies are needed to accurately determine the ocular PK of mAbs under inflammatory conditions.

Anti-TNFα intravitreal injections: Following intravitreal administration, biologic drugs are distributed from the vitreous humour to the posterior (retina) and anterior (aqueous humour) segments of the eye and are eventually eliminated by disposal into the systemic circulation [114,115]. The factors that affect the drug distribution in vitreous humour are dictated by its diffusive and convective properties through the vitreous, and the possible drug interactions with the vitreous humour elements [107].

The intravitreal administration of anti-TNFα drugs such as IFX or ADA could be a potential resource to increase its bioavailability in ocular structures, thereby achieving intraocular therapeutic concentrations by minimizing its systemic absorption and toxicity, together with decreased ADAbs generation. Hence, this possibility has been raised for the treatment of inflammatory ocular diseases. Some studies that have evaluated the effect of the intravitreal administration of IFX (dose between 1 and 1.5 mg) in patients with Behçet’s syndrome uveitis have observed an improvement in central macular thickness and visual acuity, without appreciating adverse events [116,117]. The reported improvement in clinical parameters places intravitreal IFX as a promising strategy in the treatment of ocular inflammation. However, previous results differ from recent findings of severe immunological reactions and a high percentage of therapeutic failure after intravitreal administration of IFX in patients with posterior uveitis associated with Behçet’s disease [118]. Furthermore, evidence of an inflammatory reaction and a strong suggestion of retinotoxicity to intravitreal IFX were shown in a pilot safety study of patients with diabetic macular oedema or choroidal neovascularization secondary to age-related macular degeneration who failed conventional therapies [119].

ADA (and IFX in Japan) is the only biological drug with an approved indication for the treatment of NIU, but only under systemic administration for which its safety and efficacy have been widely demonstrated. Conversely, current evidence of the safety and efficacy of intravitreal administration of ADA is scarce and contradictory. Intravitreal ADA administration has been shown to effectively improve best-corrected visual acuity, control inflammation, limit uveitis flare and decrease cystoid macular oedema in six out of seven patients with Behçet and idiopathic uveitis. One patient failed treatment but was able to regain baseline vision with no permanent effect [120]. A retrospective study evaluated the usefulness of intravitreal ADA as a rescue treatment in flares of patients receiving chronic treatment with systemic ADA. Four patients with Behçet’s disease panuveitis maintained on systemic ADA therapy prior to the panuveitis breakthrough attack were included. Of the 13 attacks documented in seven eyes, three resolved with one injection and 10 needed more than one monthly injection for resolution. This work shows that intravitreal ADA is of potential utility as a rescue therapy in patients with NIU on systemic ADA therapy that requires tighter control of the inflammation [121]. On the contrary, intravitreal ADA (monthly injections for three months) showed no efficacy in improving visual acuity or reducing central retinal thickness in eight patients with chronic uveitic macular oedema who had failed steroid treatment, although the intervention was deemed safe [122]. Evidence from preclinical studies is also inconsistent. Some studies in rabbits have reported no ocular toxicity for doses up to 5 mg of intravitreal ADA [123,124], whereas another study has reported retinal necrosis at doses of 1 mg [125]. Despite these discrepancies, a promising result of the potential utility of intravitreal administration of ADA as a strategy to increase ocular drug bioavailability has been recently obtained by García-Otero et al. [126]. These authors evaluated the pharmacokinetic profile and the biodistribution of the intravitreal administration of ^89^Zr-adalimumab in a uveitis rat model using PET imaging and showed that ADA remained about twice as long in the vitreous of diseased rats compared to unaffected ones and that its ocular permanence was around three times higher in rats with uveitis.

Further research is required to convincingly establish the efficacy and safety of intravitreal administration of ADA before considering its approval for the treatment of ocular inflammatory diseases such as NIU. The main advantage of intravitreal administration is that ocular exposure to anti-TNFα drugs is increased using lower doses than those used in systemic administration while avoiding its possible adverse effects. For instance, the systemic administration of anti-TNFα presents certain risks such as the reactivation of latent tuberculosis and in the case of IFX, it is contraindicated in patients suffering from congestive heart failure [127]. Nevertheless, intravitreal administration also has some drawbacks, mainly the invasiveness of the procedure and its possible complications.

## 5. Pharmacogenetics (PG) of Anti-TNFα in NIU

Genetic variability is one of the factors that explain the inter-individual variability in the response to treatments or in the appearance of toxicities. Unfortunately, a major limitation of PG studies on drugs used in the management of ocular inflammatory diseases is the lack of consistency due to heterogeneous study designs, different outcome measures, and small simple sizes, which possibly result in false-positive associations [128]. For this reason, PG studies focused on anti-TNFα drugs for the treatment of NIU have not been conducted. However, the treatment of NIU is very similar to that of other immune-mediated pathologies such as RA or IBD, where the influence of genetic polymorphisms on the response to biological drugs such as anti-TNFα has been evaluated [128]. Table 3 shows some of the genetic associations identified with the response to anti-TNFα agents in immune-mediated diseases that could shed light on the influence of genetics on the response to treatment in NIU. More extensive evidence of gene polymorphisms that may act as predictors of response to anti-TNFα biologic drugs in related diseases can be found in the scientific literature. 

### 5.1. Candidate Gene Association Studies

Candidate gene association studies have described different polymorphisms that can act as predictors of response to therapy with anti-TNFα agents. These polymorphisms are located mainly in genes involved in the activation of NFkB through the metabolic pathway of Toll-like receptors (TLR), genes that regulate TNFα signalling, and cytokines regulated by NFkB and involved in the metabolic pathway of helper T cells [129]. In a meta-analysis conducted by Bek et al. [130] of 47 studies that analysed the genetic differences between patients with RA responders and non-responders to anti-TNFα therapy, six polymorphisms of six different genes were found to be involved in the response: CHUK, PTPRC, TRAF1/C5, NFKBIB, FCGR2A, and IRAK3. These were genes predominantly involved in the adaptive response, unlike others located in IBD, such as those associated with TLR, which are more involved in the innate immune response. Bank et al. [131] studied 738 patients with IBD, including CD, UC, or both, from a Danish cohort and identified 19 functional polymorphisms in 14 genes associated with response to treatment with anti-TNFα agents (TLR2, TLR4, TLR9, LY96, CD14, MAP3K14, TNFA, TNFRSF1A, TNFAIP3, IL1B, IL1RN, IL6, IL17A, and IFNG) that were implicated in the inflammatory response mediated by NFκB. These associations allowed for distinguishing not only which patients would benefit from treatment with anti-TNFα agents, but also to identify those who would benefit from treatment with an agent whose target was another cytokine such as IL-1b, IL-6 or IFN- γ, or a combination of several agents [131]. Notably, these findings have recently been replicated and extended by the same authors in a different cohort of 1045 Danish patients with IBD [132]. 

The possibility of identifying patients at risk of developing immunogenicity against anti-TNFα drugs, and therefore of presenting a worse response to treatment, may guide the clinician’s choice of treatment towards the use of concomitant immunosuppressants associated with a lower incidence of ADAbs or the use of biological drugs with other targets. A positive association with the risk of developing ADAbs has been described for a polymorphism of the *CXCL12* gene, which is consistent with the well-known role of this chemokine in antibody affinity maturation and plasma cell survival required for antibody development. This genetic association was confirmed at the protein level. Elevated serum CXCL12 levels (dichotomized by the median value) were associated with a 2-fold increased risk of immunogenicity, although this analysis was restricted to RA patients treated with 4 different anti-TNFα agents [133]. Another factor that appears to be closely related to the development of immunogenicity is the level of IL-10. In patients with RA, the formation of ADAbs against ADA was associated with certain genetic polymorphisms and haplotypes of the promoter region of the *IL-10* gene [134]. Unfortunately, it was not specified how the associated polymorphisms/haplotypes influence IL-10 production. In another study in which IL-10 levels were measured in 17 patients with various immune-mediated diseases under IFX treatment, the absence or low IL-10 production and low IL-10/IFN-γ ratio were associated with an increased formation of ADAbs to IFX [135]. However, certain caution should be maintained toward these results due to the small number of patients and the heterogeneity of the diseases studied.

### 5.2. HLA Complex

A promising marker for detecting patients at higher risk of developing ADAbs, the *HLA-DQA1*05* allele, has recently been identified. Sazonovs et al. [136] found that the presence of one or two copies of the *HLA-DQA1*05* alleles conferred a 2-fold increased risk of developing immunogenicity to anti-TNFα therapy in patients with IBD, regardless of the type of anti-TNFα (ADA or IFX) or concomitant treatment with immunomodulators. This finding was replicated and extended in a multicohort prospective study of patients with multiple sclerosis (MS), RA, and IBD conducted by the European ABIRISK (Anti-Biopharmaceutical Immunization: prediction and analysis of clinical relevance to minimize the RISK) consortium [133]. This work not only confirmed the doubled risk of developing ADAbs with the presence of *HLA-DQA1*05* alleles, but also observed a 4-fold increased risk of ADAbs for patients homozygous for these alleles. Among the clinical factors evaluated, concomitant immunosuppressant treatment reduced the risk of immunogenicity, in contrast to findings from the previous study, whereas tobacco consumption showed a positive association with ADAbs development.

It has been suggested that ADAbs development against biological drugs may share common immunogenetic pathways across diseases in view of the similarities shared in dynamics of antibody production and rate of immunogenicity [133]. Since immunogenicity has a great impact on anti-TNFα drug levels, PG studies in other immune-mediated diseases could provide clues as to which genetic factors contribute to the development of ADAbs in NIU, as well as to reveal genetic factors that contribute to other pathways involved in the response to anti-TNFα treatment in these patients. Stronger evidence of the influence of PG on the response to anti-TNFα treatment in NIU can be obtained from PG studies specifically conducted in patients with this pathology. Furthermore, considering all the data shown about PG in anti-TNFα therapy, and the numerous genes involved in their response, the implementation of a genomic array encompassing all the genes involved, analogous to some used in cancer therapy [137], could be of help in interpreting PG results.

**Table 3 pharmaceutics-15-00766-t003:** Association between SNPs and treatment response to anti-TNFα drugs in immune-related diseases other than NIU.

Authors [Ref.]	Gene	SNP (Allele)	Effect of the SNP	Disease	Proposed Gene/Protein Function
Bek et al. 2017 [130]	*CHUK*	rs11591741 (C)	non-response	RA	Component of a cytokine-activated protein complex that inhibits NFκB.
	*PTPRC*	rs10919563 (A)	non-response	RA	Suppresses JAK kinases, functions as a regulator of cytokine receptor signalling.
	*TRAF1/C5*	rs3761847 (G)	non-response	RA	Required for TNFα-mediated activation of MAPK8/JNK and NFκB. Mediates the anti-apoptotic signals from TNF receptors.
	*NFΚBIB*	rs9403 (C)	non-response	RA	Inhibits NFκB by complexing with and trapping it in the cytoplasm.
	*FCGR2A*	rs1801274 (G)	non-response	RA	Involved in the process of phagocytosis and clearing of immune complexes.
	*IRAK3*	rs11541076 (T)	non-response	RA	Negative regulator of Toll-like receptor signalling.
Bartelds et al. 2009 [134]	*IL10*	rs6703630, rs1800896, rs1800871 (AGC haplotype)	non-response ^a^	RA	Pleiotropic cytokine with a role in immunoregulation and inflammation, enhances B cell survival, proliferation, and antibody production, can block NFκB activity, and is involved in the regulation of the JAK-STAT signalling pathway
		rs6703630, rs1800896, rs1800871 (GAT haplotype)	response ^a^		
Bank et al. 2014 [131]	*TLR2*	rs4696480 (T)	non-response	Only UC	Activates inflammation through the canonical NFκB pathway.
		rs11938228 (A) ^b^	non-response	IBD	
		rs1816702 (T)	response	Only CD	
		rs3804099 (C)	response	IBD	
	*TLR4*	rs1554973 (C) ^b^	non-response	IBD	Activates inflammation through the canonical or noncanonical NFκB pathway.
		rs5030728 (A) ^b^	response		
	*TLR9*	rs352139 (A)	non-response	IBD	Activates inflammation through the canonical NFκB pathway.
		rs187084 (C)	response		
	*CD14*	rs2569190 (A)	non-response	Only UC	Binds LPS and transport it to TLR4
	*TNFA*	rs361525 (A)	non-response	IBD	Pro-inflammatory cytokine activated by NFκB1.
	*TNFAIP3*	rs6927172 (G)	non-response	IBD	Inhibits NFκB activation and TNFα-mediated apoptosis.
	*IL1RN*	rs4251961 (C) ^b^	non-response	Only UC	Inhibits IL-1β signalling.
	*IL17A*	rs2275913 (A)	non-response	IBD	Pro-inflammatory cytokine activated by NFκB1, induces production of IL-1β, IL-6, and TNFα.
	*LY96*	rs11465996 (G)	response	IBD	Binds to TLR2 or TLR4 and is required for their activation to LPS stimuli
	*MAP3K14*	rs7222094 (C)	response	IBD	Central kinase in the noncanonical NFκB pathway
	*TNFRSF1A*	rs4149570 (T) ^b^	response	IBD	Binds TNFα and initiates a kinase cascade.
	*IL1B*	rs4848306 (A)	response	IBD	Pro-inflammatory cytokine activated by NFκB1.
	*IL6*	rs10499563 (C)	response	IBD	Pro- and anti-inflammatory cytokine activated by NFκB1.
	*IFNG*	rs2430561 (A)	response	IBD	Pro- and anti-inflammatory cytokine activated by NFκB1.
Bank et al. 2019 [132]	*NLRP3*	rs4612666 (T)	non-response	IBD	Member of the NLRP3 inflammasome complex, upstream activator of NFκB signalling.
	*IL18*	rs187238 (C)	response	Only CD	Proinflammatory cytokine of the IL-1 family, capable of stimulating IFNγ production.
		rs1946518 (T)	response	IBD	
	*JAK2*	rs12343867 (C)	response	IBD	Plays a central role in cytokine and growth factor signalling, downstream target of IL6
	*NFΚBIA*	rs696 (A)	response	IBD	Complexes with REL dimers inhibit NFκB/REL complexes.
Hässler et al. 2020 [133]	*CXCL12*	rs10508884 (T)	non-response ^a^	Several diseases	Plays a role in embryogenesis, immune surveillance, antibody affinity maturation, inflammation response, tissue homeostasis, and tumour growth and metastasis.
	*HLA-DQ*	*HLA-DQA1*05*	non-response ^a^		Plays a central role in the immune system by presenting peptides from extracellular proteins.
Sazonovs et al. 2020 [136]	*HLA-DQ*	*HLA-DQA1*05*	non-response ^a^	CD	Plays a central role in the immune system by presenting peptides from extracellular proteins.

^a^ The genetic association with the response was indirectly ascertained through the association of the SNP with the development of ADAbs, which has been related to a worse response to anti-TNFα agents. Therefore, response represents that the SNP is negatively associated with ADAbs development, whereas non-response represents a positive association of the SNP with ADAbs development. ^b^ Association replicated in Bank et al. [132]. SNP: Single Nucleotide Polymorphism, RA: rheumatoid arthritis, UC: ulcerative colitis, IBD: intestinal bowel disease, CD: Crohn’s disease.

## 6. Conclusions

The implementation of TDM of biological drugs in the field of NIU could help to optimize treatments and obtain better response rates, as already shown in other immune-mediated diseases. The clinical benefits of effective anti-TNFα treatment in NIU include better control of ocular inflammation, decrease in the number of flares, reduction in visual loss, and improvement in the quality of life for patients. However, anti-TNFα treatment can also result in adverse events, especially when supratherapeutic levels are present. Infections, congestive heart failure, demyelinating diseases, drug-induced systemic lupus erythematosus or induction of psoriasis are some of the potential adverse events associated with anti-TNFα therapy. Although the measurement of anti-TNFα and ADAbs levels is not routinely used in clinical practice in NIU, it has been shown to be useful in a series of non-randomized observational studies in patients with refractory NIU. This highlights the critical need for clinical studies to convincingly establish the usefulness and cost-effectiveness of TDM-based strategies, over empirical dose escalation strategies to guide treatment adjustment with biological drugs in the treatment of NIU and define a specific therapeutic range. Although the influence of genetic polymorphisms on the response to biological drugs has been barely explored in NIU to date, PG may be an important aspect in optimizing and predicting response to biological treatments and influencing TDM, as has been evidenced in other immune-mediated diseases. At the preclinical level, studies should further address the degree of distribution of therapeutic proteins in the eye under inflammatory conditions, in order to improve knowledge about the biopharmaceutical behaviour of mAbs in this disease. 

## Figures and Tables

**Figure 1 pharmaceutics-15-00766-f001:**
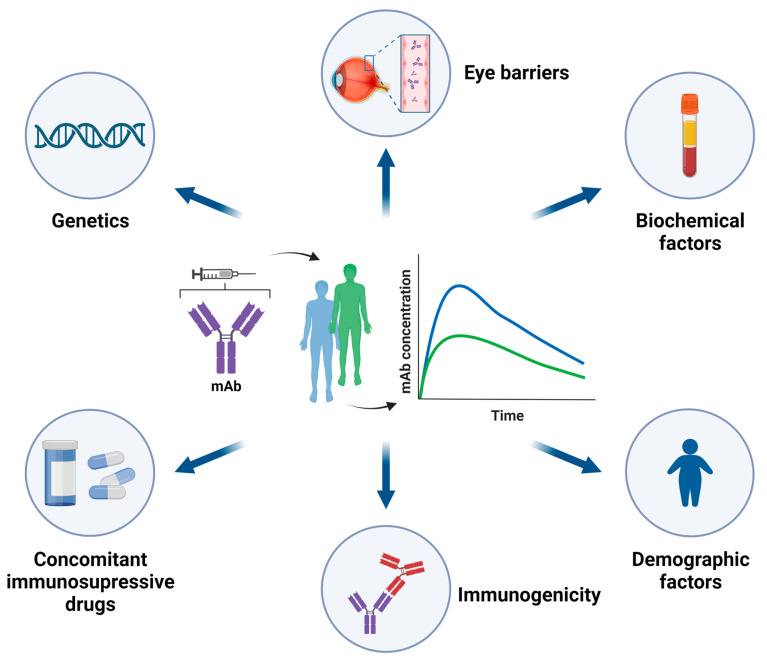
Factors that influence the pharmacokinetics of biologic agents in NIU. Created with biorender.com.

**Figure 2 pharmaceutics-15-00766-f002:**
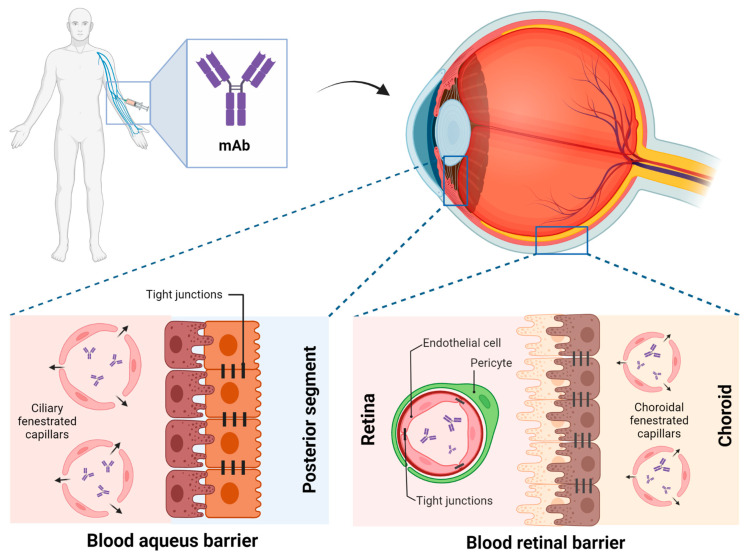
Ocular barriers encountered by biologic drugs to reach their ocular therapeutic target after systemic administration. Created by biorender.com.

## Data Availability

No new data were created or analyzed in this study. Data sharing is not applicable to this article.

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
