# Peer review of "Perspectives of Therapeutic Drug Monitoring of Biological Agents in Non-Infectious Uveitis Treatment: A Review"

_pharmaceutics, 2023, doi:10.3390/pharmaceutics15030766_

Round 1

Reviewer 1 Report (New Reviewer)

Title 

The current title doesn´t describe adequately the type of work presented. “Perspectives of Therapeutic Drug Monitoring of Biological Agents in Non-Infectious Uveitis Treatment. A Review” or similar would be more adequate. As the authors describe, not all the agents mentioned are monoclonal antibodies (nor are all of them anti-TNFα). On the other hand, TDM resumes pharmacokinetics and pharmacogenetics. A concise and concrete title is to be welcomed.

Abstract

Lines 29-30. I suggest, “Therapeutic drug monitoring (TDM) of drug and anti-drug antibody (ADAbs) levels is emerging as a resource to optimize biologic therapy by personalizing treatment to bring and maintain drug concentration within the therapeutic range”, because the therapeutic effect is mostly dependent on an adequate exposure, not only on a Cmax.

Keywords

Some of them are not MeSH terms (non-infectious uveitis (NIU), immune-mediated disease, anti-tumour necrosis factor α (anti-TNFα)). Non-infectious uveitis (NIU) is relevant in the context analyzed, but the others could be omitted without problem.

Introduction

It summarizes adequately the current knowledge about pathophysiology and therapeutic of NIU. However, the two initial paragraphs of “3. Therapeutic Drug Monitoring of Anti-TNFα in NIU” should be rewritten and incorporated to the Introduction, because are where the need for TDM and its possible/sure contribution in this disease is based, beyond the general contributions of the TDM reflected in the current version of the draft. I suggest moving them to the lines 126-127. In line 54, “chronic doses of prednisone equivalents over 7.5 mg” per day has to be added. In line 102, mAb as acronym of monoclonal antibodies has to be introduced after “The passage of monoclonal antibodies” for being the first mention and further used in the rest of the text. “TDM was typically considered advantageous for drugs with a large inter-individual variability in exposure with relatively low intra-individual variation, a significant exposure–efficacy relationship, a narrow therapeutic window, and availability of a validated bioanalytical assay. It has been postulated recently that this could also represent a useful tool to individualize dosing and optimize treatment using drugs with a wide therapeutic window and high cost” (Escudero-Ortiz, V et al. Relevance of Therapeutic Drug Monitoring of Tyrosine Kinase Inhibitors in Routine Clinical Practice: A Pilot Study. Pharmaceutics 2022, 14, 1216.). This cite can help to more precisely argue about the convenience of using TDM in mAb therapeutics in addition to the arguments exposed in the draft paragraph.

2. Biologics in uveitis treatment

This section has to be reformulated. The description of every drug has to be uniform, including in the text relevant characteristics as type of drug or target and mechanism of action. The current version submits initially to Table 1 and exposes in the body of the text a non-organized mixture of data about every drug. I suggest to suppress Table 1 and to incorporate corresponding data in the text in a uniform and organized manner.

In line 212, there is no previous explanation for the acronym VKH.

3. Therapeutic Drug Monitoring of Anti-TNFα in NIU

In line 274 etanercept is listed as mAb and it´s not. Can be mentioned as a molecule similar to mAbs for example.

In lines 274-275, the sentence “Clearance of ADA and IFX is significantly increased in men than in women” is better expressed as “Clearance of ADA and IFX is significantly higher in men than in women”. Sex as a factor influencing clearance doesn´t change.

In lines 276-278, the sentence “Despite the direct relationship between ADA and IFX clearance with body size, ADA dosage is not adjusted to weight for the treatment of NIU, whereas IFX doses are weight-adjusted.” has to be explained. Refer the reason for this apparently illogical circumstance and mention it even if there is no scientific reason.

In lines 281-282 there is a sentence without verb: “In inflammatory bowel disease (IBD), an inverse relationship between serum albumin levels and clearance of ADA [60] and IFX [61]. Please correct it.

3.1.4. Concomitant immunosuppressive therapy

Consider rewrite this section. In lines 304-312 the superiority of combined therapy seems a proven fact. In lines 313-327, a possible explanation of the benefit of concomitant administration of immunosuppressants and mAbs is formulated, but repeating arguments in the text. Finally, the section finishes with negative data. A new text, starting with positive and negative data followed by the hypothesis formulated in a succinct and organized manner, would express the message much more clearly.

3.2. Evidence supporting TDM of Anti-TNFα in NIU

“Although these preliminary data supporting the monitoring of anti-TNFα drug levels and ADAbs to optimize NIU treatment are promising and are in line with observations in other pathologies [45,80–83], they are insufficient to implement TDM in routine clinical practice.” Where this last conclusion come from?

“Therefore, more studies are needed aimed to evaluate whether guiding treatment using TDM-based strategy leads to greater clinical benefits and cost savings than empirical dose escalation of anti-TNFα drugs in NIU, an approach that has proven safe and cost-effective compared to no treatment optimization [86].” Dose escalation assumes a strategy with a significant number of underdosed cases in initial therapy. Whenever it has been used, TDM added precision against empiricism, avoiding futility by underdosing and toxicity (including financial one) by overdosing in a faster way than dose escalation.

Finally, data of some guidelines aiming to use TDM in some cases of NIU are described.

The last part of the section, at least since line 387, has to be rewritten, presenting data (pros and cons) followed by the interpretations and conclusions in an organized manner,

Consider to rewrite 3.1, 3.2 y 3.4 as a unique section about TDM in NIU. There are many nuances that do not add relevant information and the subject can be exposed in a more attractive and informative way. The lack of organization and the profusion of data make current version sometimes confuse.

Anti-TNFα intravitreal injections:

In line 572, “…by minimizing its systemic absorption and toxicity.”, consider to add the possibility of decreased anti-mAb generation.

 5. Pharmacogenetics (PG) of Anti-TNFα in NIU

“However, clear evidence of the influence of PG on the response to anti-TNFα treatment in NIU will only be obtained from PG studies specifically conducted in patients with this pathology.” This statement is difficult to assume in its current formulation. The previous part of the paragraph indicates the opposite. Consider something like “stronger evidence of the influence of PG on the response to anti-TNFα treatment in NIU can be obtained from PG studies specifically conducted in patients with this pathology”.

Considering all the data shown about PG in anti-TNF agent’s therapy, the implementation of a genomic array that encompasses all of them, analogous to some employed in cancer therapy, can be suggested.

Conclusions

“Although the measurement of anti-TNFα and ADAbs levels is not routinely used in clinical practice in NIU, it has been shown to be useful in series of non-randomized observational studies in patients with refractory NIU. This highlights the critical need for clinical studies to convincingly establish the usefulness and cost-effectiveness of TDM-based strategies, over empirical dose escalation strategies to guide treatment adjustment with biological drugs in the treatment of NIU and define a specific therapeutic range.” Already commented in section 3.2.

“Furthermore, PG may be an important aspect in optimizing and predicting response to biological treatments, although this field has been barely explored in NIU to date.” The paper is about TDM and PG has to be presented, even in conclusions, as a factor influencing TDM. Otherwise, it appears as a questionable addition.

Author Response

Reviewer 2 Report (New Reviewer)

In this manuscript by Busto-Iglesias M. and colleagues, the authors have nicely reviewed current literature on Pharmacokinetics, Therapeutic Drug Monitoring and Pharmacogenetics of Monoclonal Antibodies in Non-Infectious Uveitis Treatment.

In their work, the authors have well organized and sufficiently described the main aspects of the choosen Topic,. However, in my opinion, there are some concerns that should be addressed:

-At 3.1 section, the authors describe the Pharmacokinetics (PK) of monoclonal antibodies. In this section, a hint on the "antigen mass" as source of inter-individual  PK variability should be included.

-Similarly, presence of comorbidities such as diabetes has been related to the increased clearance of mAb. Is non-infectious uveitis often associated to concomitant pathologies that could affect mAb elimination rate?

-In my opinion, the authors should include a section/paragraph/table describing the assays actually used for measuring both drug and anti-drug antibodies concentrations. Moreover, for each assay a list of main advantages and disadvantages should be reported;

-In TDM section, the authors should include considerations on the potential cost-effectiveness of proactive versus reactive TDM of Adalimumab in patients with NIU.

- A section on the safety profile and the delayed adverse events reported during NIU treatment with biologics should be added. Perhaps, a table with the main causes of discontinuation could be considered.

- Language should be deeply revised throughout the whole manuscript. Some sentences are too long and the meaning is lost (i. e lines 89-97; 98-102; 304-306).

Specific points:

Line 131: Replace "with" with "to"

Line 263: Remove "The"

Line 266: Avoid repetition

Line 276: Replace "due to" with "by".

Line 309: Remove "This"

Line 313: Remove "The"

Line 390: Remove aimed

Line 535: Remove "a"

Line 720: Remove "has been"

Line 723: Replace "of" with "for".

Author Response

Reviewer 3 Report (New Reviewer)

The authors tried to review the available evidence on the PK, TDM and PG monitoring of anti-TNFα drugs used in NIU, as well as to discuss the relevance of the biopharmaceutical considerations that concern drug delivery in the eye, in relation with biological drugs treatment. This is a contemporary topic, considering the potential benefits of biologics TDM in inflammatory diseases, but the contradictory findings regarding its usefulness. I think the whole article is positive considering TDM, but with very less opposite findings. I gave some comments and suggestions that should be addressed

“Corticoids” should be replace with “corticosteroids”

“Therapeutic outcome of biologic drugs in immune mediated diseases is closely related to serum drug concentration.” This sentence need at least one reference.

PK of mabs was given descriptively, it should be follow with some numbers especially there is few mabs approved for NIU treatment. You provides some information later in the manuscript, but as reader I prefer to be at the beginning where you talk about mabs pharmacokinetics.

 ,,C-reactive protein (CRP) also influences the PK of IFX. Specifically, a direct correlation has been described between CRP levels and IFX clearance [62].” Do you have potential explanation for this phenomena?

 3.1.4. Concomitant immunosupressive therapy

Authors constantly said ,,immunosuppressants”, but I think it should be concrete, which immunosuppressant or if the findings they cited are related to all of them, it should be stated. It is important for the further research and more important for clinical practice. From the references, I can see methotrexate.

Same with immunomodulators.

As you cited literature date that support the TDM of biologics, you should also cited authors that oppose this findings.

English should be checked to be more attractive to the readers.

Author Response

Reviewer 4 Report (New Reviewer)

The paper entitled “Pharmacokinetics, Therapeutic Drug Monitoring and Pharmacogenetics of Monoclonal Antibodies in Non-Infectious Uveitis Treatment” is a review based therapeutic drug monitoring (TDM) in Non-Infectious Uveitis (NIU) and in other immune-mediated diseases and how TDM and pharmacogenetics can be a useful tool in managing these patients.

The review shows how the implementation of TDM of biological drugs in the field of NIU could help to optimize treatments and obtain better response rates. The review also provides a description of the importance of the clinical benefits of effective anti-TNFα treatment in NIU and how best to manage patients undergoing this treatment protocol.

The paper is thorough and highlights the important issues behind inflammation and underlying pathways in NIU. The study adds to the literature. The use of headings and subheadings gives the paper structure and a logical organization of specific correlated topics.   

Minor editing can improve the English and flow of the text.

The study has been correctly planned and represents a solid basis for future studies regarding potential novel targets for diagnosis and treatment. It is nicely written and of clinical interest. References are appropriate. The figures and table are pertinent, and descriptive and assist in describing the results. 

Round 2

Reviewer 1 Report (New Reviewer)

Nothing to consider. The new draft accomplishes with the observations formulated in the 1st review. Can be published.

Reviewer 3 Report (New Reviewer)

No further comments.

This manuscript is a resubmission of an earlier submission. The following is a list of the peer review reports and author responses from that submission.

Round 1

Reviewer 1 Report

The authors of the publication prepared a review on the monitored therapy of uveitis with the use of biotechnological drugs. Therapeutic Drug Monitoring (TDM) is not a new procedure. In the case of classic drugs, such as digoxin or carbamazepine, we already have over 20 years of observational studies that preaty summarize the effects of TDM in these patient groups. First publications appear in the case of biotechnological drugs, including antibodies.
A very interesting manuscript, but to increase its impact, please:
- in the introduction, please compare the ideas of TDM in the group of antibodies with classic drugs.
- please give your opinion on the differences in interpretation of the obtained results
- The conclusion is too general. In other words, you can summarize the manuscript in this way, but before that, please specify clearly (e.g. at the end of the description of each drug) what are the benefits of TDM for a specific drug, e.g. what side effects, what effects can be modulated? It is extremely important because the recipients of the publication are not only pharmacists but also rheumatologists, immunologists and clinical pharmacologists. For them, the clinical summary of TDM is extremely important.

Reviewer 2 Report

The study written by Manuel Busto-Iglesias et al. concerns the pharmacokinetic and pharmacogenetic aspects in the treatment of non-infectious uveitis (NIU).  The authors collected current knowledge on the Biological drugs targeting different cytokines ( TNFα, IL-6, IL-17 or IL-23) which play a key role in NIU inflammatory process.

The introductory part should be specified: what kind of review is it, what period does the collected literature cover, and what databases were searched to select the discussed and cited research papers. At the end of the Introduction, authors should present a short outline of individual chapters to familiarize the reader with the scope of the work,

line 108-The authors wrote: "the biopharmaceutical characteristics of the eye" - what exactly did they mean? Biopharmaceutical considerations rather concern drug delivery. Please explain or correct.

Table 2, 3 – instead of Authors please give the number of appropriate references.

Line 187 Authors wrote the following “A growing number of reports have supported the use of rituximab in some types of NIU.”- Please give appropriate references.

Line 120-192- The information in the above section is actually in Table 1. I suggest removing most sentences covering data from Table 1.

Fig.1-In my opinion, this figure such be thought over to be useful for readers.

Reviewer 3 Report

The issue described by the authors is an innovative approach to therapy with biological drugs, especially in ophthalmology. So far, drug concentration-monitored therapy has been most often used to personalize traditional drugs, and there are few reports on the use of this method in biological treatment. Additionally, the linking of TDM with pharmacogenetic studies is an added value. The presented topic is part of the modern idea of personalized therapy and the authors have explored this issue well using various techniques in the form of tables and figures. The literature is adequate for the topic.

Reviewer 4 Report

This review provide relevant information about mAbs PK/PD in non-infectious uveitis treatment.

As TDM interest is largy mentionned, title  could be change in: PK/PD and TDM of mAbs in NIUT.

Round 2

Reviewer 1 Report

after introducing corrections, the paper may be considered for publication